# Size Exclusion Chromatography–Mass Photometry: A New Method for Adeno-Associated Virus Product Characterization

**DOI:** 10.3390/cells12182264

**Published:** 2023-09-13

**Authors:** Di Wu, Xiaonan Zhao, Diego Antonio Jimenez, Grzegorz Piszczek

**Affiliations:** 1Biophysics Core Facility, National Heart, Lung and Blood Institute, National Institutes of Health, Bethesda, MD 20892, USA; piszczeg@nhlbi.nih.gov; 2Laboratory of Cell and Molecular Biology, National Institute of Diabetes, Digestive and Kidney Diseases, National Institutes of Health, Bethesda, MD 20892, USA; xiaonan.zhao@nih.gov (X.Z.); diego.jimenez@pennmedicine.upenn.edu (D.A.J.)

**Keywords:** AAV titer, Vg/Cp ratio, sample quality, gene therapy

## Abstract

Over the past decade, adeno-associated viruses (AAVs) have attained significant prominence in gene therapy and genome editing applications, necessitating the development of robust and precise methodologies to ensure the quality and purity of AAV products. Existing AAV characterization techniques have proven effective for the analysis of pure and homogeneous AAV samples. However, there is still a demand for a rapid and low-sample-consumption method suitable for the characterization of lower purity or heterogeneous AAV samples commonly encountered in AAV products. Addressing this challenge, we propose the SEC-MP method, which combines size exclusion chromatography (SEC) with mass photometry (MP). In this novel approach, SEC effectively separates monomeric AAV particles from impurities, while the UV detector determines the virus particle concentration. MP complements this process by estimating the fraction of fully packaged AAVs in the total population of AAV particles. This combined methodology enables accurate determination of the titer of effective, fully packaged AAVs in samples containing aggregates, incorrectly packaged AAVs with incomplete genomes, protein or DNA fragments, and other impurities. Our experimental results demonstrate that SEC-MP provides valuable guidance for sample quality control and subsequent applications in the field of AAV research.

## 1. Introduction

Adeno-associated viruses (AAVs) have emerged as a leading choice for gene therapy vectors, with hundreds of clinical trials utilizing AAVs as a gene delivery tool due to their low toxicity, replication defective nature, non-integrating properties, and ability to infect both dividing and non-dividing cells [1,2,3]. Due to these favorable properties, AAV utilization in the industry has experienced rapid growth, leading to an increasing demand for AAV product characterization and quality control. The FDA has outlined specific parameters that should be included in the application for human gene therapy investigational new drugs, including the ratio of infectious to non-infectious particles or full to empty particles [4]. To address the growing demand for AAV product characterization, a range of methods and technologies based on both mature and emerging techniques are being developed. These methodologies aim to address the challenges of AAV production, including the need for rigorous quality control measures.

Recombinant AAV vectors are small viruses with a diameter of approximately 26 nm, in which the replication and encapsulation genes are replaced by the gene of interest (GOI). During the assembly process, some virions may undergo improper assembly, leading to the formation of defective particles, including empty capsids or vectors containing incomplete portions of the genome (referred to as partially packaged AAVs) [5,6]. While the clinical impact of incorrectly assembled virions in gene therapy outcomes is debated [7,8,9], it remains important to accurately characterize the viral product and estimate the fraction of effective, fully packaged AAV particles reliably and precisely.

The characterization of AAV poses several challenges. Due to the identical geometric size of correctly and incorrectly packaged AAV particles, molecular diffusion-based technologies like dynamic light scattering and size exclusion chromatography can detect aggregates or small molecule contaminants in AAV samples, but they fail to discriminate between monomeric AAVs containing correct and incorrect DNA cargos. Technologies that measure the average mass of all species in the sample, such as static light scattering, or that observe the virion morphology, such as cryo-electron microscopy, are not able to accurately quantify the partially packaged AAVs [10]. While qPCR can directly quantify the copy number of the GOI in the sample, and the combination of qPCR and ELISA can be used to determine the percentage of fully packaged AAVs (%full) in samples [11], they may overlook certain impurities such as structurally deficient capsids, and have limitations in the precision of titer determination due to variations in amplification efficiency and standard curve accuracy [12]. New technologies, such as droplet digital PCR (ddPCR), offer improved measurement accuracy of AAV titer determination, especially with self-complementary AAVs, as they eliminate the need for a standard curve and are less sensitive to amplification efficiency. However, ddPCR introduces additional complex workflow steps [13,14], and does not eliminate primer-dependency.

Recent advancements involving the combination of technologies, such as SEC-MALS [15] (size exclusion chromatography and multi-angle static light scattering), IEC-MALS [16] (ionic exchange chromatography and static light scattering), and Stunner [17] (UV absorbance, dynamic light scattering, and static light scattering), have shown promise in successfully quantifying both the titer and %full of AAV samples. However, it is important to note that these methods are most effective for samples consisting exclusively of highly homogeneous empty and fully packaged AAVs without any partially packaged species. In practical scenarios involving long-term sample storage, freeze–thaw cycles, or AAVs containing heterogenous genomes, the composition of AAV samples becomes more complex, greatly limiting the applicability of the aforementioned methods in characterizing such samples.

Over the past decade, analytical ultracentrifugation (AUC) has emerged as the gold standard for AAV sample quality assessment. AUC provides a high-resolution differentiation of variably packaged AAVs, relying on their distinct buoyant masses for characterization. Moreover, by employing light interference and absorbance detection at various wavelengths, AUC provides detailed information regarding the protein and DNA content within the virus [18]. However, limitations such as the need for substantial material quantities and the long experimental time hinder the widespread use of AUC for AAV sample characterization [19].

Recently, the single-molecule technology, mass photometry (MP), has been successfully applied for AAV characterization [20]. This study demonstrates that the mass distributions obtained from MP have resolution comparable to the sedimentation coefficient distributions obtained from AUC, particularly for heterogenous AAV samples containing multiple partially packaged species. Notably, a typical MP measurement only requires sub-picomolar amounts of a sample, and data acquisition and analysis can be completed within 5 min. The amount of material and experimental time required by MP correspond to approximately 1/600 of the sample amount and 1/40 of the experiment time needed for AUC. The main limitation in utilizing MP for AAV characterization is the absence of titer information from the MP data. This is particularly significant since the titer will determine the AAV dosage for subsequent studies.

To provide a comprehensive AAV characterization, the current study combines MP with size exclusion chromatography (SEC-MP). In the SEC-MP method, the UV detector of the SEC system is utilized to determine the number concentration (molar concentration) of the total population of monomeric AAV particles and the total amount of encapsidated DNA. The implementation of SEC plays a pivotal role in isolating monomeric AAV particles from impurities and aggregates, thereby enabling a focused and selective analysis of the monomeric AAV subpopulation. Consequently, the UV absorbance signals recorded by the UV detector exclusively characterize the proteins and DNAs present in the monomeric AAVs, leading to concentration calculations that accurately reflect the average protein and DNA content within the intact virus. It is worth noting that the UV extinction coefficient for DNA remains independent of its sequence, and the extinction coefficients for capsid proteins only exhibit slight variations across different serotypes [19]. Consequently, the same extinction coefficients can be used for different AAV serotypes, simplifying the SEC-MP method application.

Potential variations in the masses of monomeric AAVs within the sample may occur due to the presence of incorrectly packaged AAVs. The SEC-MP method incorporates MP, which enables an investigation into the mass distribution of monomeric AAVs within the sample. This facilitates the direct estimation of the fraction of fully packaged AAVs (or other subpopulations) through the analysis of the MP mass distributions. By combining this information with the total number concentration of all monomeric AAVs provided by UV analysis, the titer of effectively, fully packaged AAVs can be estimated. This also provides valuable insights into the presence of incorrectly assembled AAVs and impurities.

In this study, high-genome-purity and low-genome-purity AAV samples in the presence of different amounts of impurities are evaluated via the SEC-MP method, and the results are compared with theoretical values and qPCR measurements. Our results suggest that SEC-MP is a promising method for the fast estimation of effective AAV titers in complex and heterogeneous samples.

## 2. Materials and Methods

### 2.1. Size Exclusion Chromatography (SEC)

SEC experiments were performed using the Agilent 1100 series HPLC system including an isocratic pump, autosampler, and diode array detector (Santa Clara, CA, USA). SEC columns were obtained from Wyatt Technology (Santa Barbara, CA, USA) and the flowrates were 0.5 mL/min for the 7.8 × 300 mm column (WTC-050N5) and 0.3 mL/min for the 4.6 × 300 mm column (WTC-050S5), respectively. Phosphate buffer at pH 7.4 with 350 mM NaCl and 0.001% Pluronic F-68 (24040032, ThermoFisher, Waltham, MA, USA) was used as the mobile phase.

### 2.2. Mass Photometry (MP)

MP experiments were performed at room temperature using a OneMP mass photometer (Refeyn, Oxford, UK) following the standard protocol [21]. AAV samples with varying sample qualities specified for the needs of this study were obtained from AAVnergene (Rockville, MD, USA). Samples directly from the AAV stocks, or collected after SEC, were measured using MP. All samples were diluted to approximately 10^11^ particles/mL and 10 µL of each sample was loaded on an MP coverslip [20]. This concentration yielded optimal measurement conditions for MP, effectively addressing the concern that an excessive sample concentration could compromise the quality of the MP data, while an insufficient sample concentration might have resulted in an inadequate dataset for constructing the mass distribution. Data were collected using AcquireMP software (Version 2023R1, Refeyn, UK) under the AAV mode, and analyzed using DiscoverMP software (Version 2023R1, Refeyn, UK).

### 2.3. SEC-MP Data Analysis

The chromatograms of absorbance signals at 260 nm and 280 nm were exported and plotted, and the baselines of both chromatograms were set manually (Figure 1A). Following the baseline subtraction, the peak boundaries corresponding to the monomer AAVs were set manually (Figure 1B). The total absorbance at 280 nm and 260 nm within the peak boundaries was integrated for the concentration calculation. Additionally, sample heterogeneity was assessed using MP.

The MP analysis of AAV sample quality determines one of 3 possible scenarios:If MP indicates that the AAV sample is highly homogeneous (pure-genome AAV) and only one narrow peak corresponding to monomeric AAVs is observed, the titer and genome size of the AAV can be directly calculated based on chromatogram analysis. The capsid molar concentration, *c*_cap_, and the DNA *w/v* concentration, *w*_DNA_, can be calculated using the following equations:
*A*_280_ = *ε*_cap280_ × *c*_cap_ + *ε*_DNA280_ × *w*_DNA_

*A*_260_ = *ε*_cap260_ × *c*_cap_ + *ε*_DNA260_ × *w*_DNA_(1)
where *A*_280_ and *A*_260_ represent the total absorbances at 280 nm and 260 nm, respectively, integrated from chromatogram peaks, as described above, while *ε* represents the extinction coefficients of the capsid (cap) and DNA at 280 nm or 260 nm. In this scenario, the number concentration of total capsids can be calculated as *N*_A_·*c*_cap_, where *N*_A_ represents the Avogadro number. The molecular mass of the encapsidated DNA is calculated as *w*_DNA_/*c*_cap_, and finally, the DNA molecular mass is converted to the genome size in kilo-nucleotide (knt) using the following formula [22]:*M.W. of ssDNA* = (# *nucleotides* × 303.7) + 79.0If MP indicates that the sample is a mixture of empty and fully packaged pure-genome AAVs, chromatogram analysis can also provide the total number concentration of capsid using Equation (1). The molecular mass of the genome, *M*_genome,_ can be obtained either from the known vector sequence or from the virion mass measured via MP. The vector genome (Vg) to total capsid (Cp) ratios, or the percentages of fully packaged AAVs (%full), can be calculated as *w*_DNA_/(*c*_cap_·*M*_genome_). The titer of an effective, fully packaged AAV is determined by multiplying the total capsid number concentration by the %full value, as *c*_cap_·*N*_A_·%full.If MP indicates the presence of partially packaged species in the sample, the chromatogram analysis can still provide the total number concentration of capsids (*c*_cap_) using Equation (1). Additionally, by analyzing the MP distribution and quantifying the total number of AAV particles (*n*_total_) along with the number of AAV particles with the correct mass detected via MP (*n*_full_), the value of %full can be determined as *n*_full_/*n*_total_. Consequently, the titer of effective, fully packaged AAVs can be calculated as *c*_cap_
*N*_A_·%full.

The chromatogram analysis was conducted using MATLAB (Version R2022b) scripts developed by the authors, and an example of the results is shown in Figure 1C. The analysis of AAV fractions was performed using the MP software DiscoverMP (Version 2023R1, Refeyn, UK). It is important to note that the presence of highly homogeneous, or pure-genome, AAVs, as indicated via MP, does not imply the absence of contaminants in the sample. Instead, it only signifies the homogeneity of the monomeric fraction of the AAV sample within the 3 MDa to 6 MDa mass range.

### 2.4. DNase-qPCR

AAVs were digested by DNaseI (New England BioLabs, Ipswich, MA, USA) prior to real-time qPCR [23] analysis. Briefly, a total of 2 × 10^9^ GC/mL of AAV (based on the concentrations provided by the vendor) was digested in 1× New England Biolabs DNase reaction buffer (New England BioLabs, Ipswich, MA, USA) with 2 U of DNaseI in 20 µL. The digestion mixtures were incubated at 37 °C for 15 min, followed by heat inactivation of DNase I at 75 °C for 10 min. The absolute copy number of the AAV genome was determined using a standard curve, with a linearized plasmid as the qPCR standard. Primers were designed to recognize the GOI. Both the linearized plasmid DNA and AAV samples were diluted in Dubelcco’s phosphate-buffered saline (DPBS) (ThermoFisher Scientific, Waltham, MA, USA). Real-time PCR was performed in quadruplicate using the PowerUp SYBR Green Master mix (Thermo Fisher Scientific) in a StepOnePlus Real-Time PCR System (Thermo Fisher Scientific), following the manufacturer’s protocol.

## 3. Results

### 3.1. SEC-MP of Single-Species AAV Samples

In the SEC-MP method, MP settings are optimized for the analysis of AAVs in the range from 3 MDa to 6 Mda, disregarding impurities outside of this range. When MP results indicate that the AAV samples are homogenous with a standard deviation (σ) of experimental mass smaller than 150 kDa as obtained from the Gaussian fit of the distribution peak, the titer and the genome size of the encapsidated DNA could be directly determined via the chromatogram analysis described in scenario 1. The genome size calculated from chromatogram analysis can be confirmed by the MP, and further compared with the theoretical values to verify the correct assembly of the AAV.

As shown via the absorbance chromatograms in Figure 2, peaks at 8–9 mL correspond to the monomeric AAVs. Figure 2A shows the results obtained for a highly homogeneous, empty AAV sample. The mass obtained from MP is 3.76 MDa with σ = 135 kDa, and the genome size calculated from the chromatogram analysis is 0. The number concentration of total capsids was determined as 2.40 × 10^12^ Cp/mL, which is lower than the supplier-provided value of 4.5 × 10^12^ Cp/mL. This could be due to the significant amount of AAV aggregates and small molecule impurities in the sample that contribute to the UV signal of the unfractionated sample (Figure 1 and Appendix A).

Another example of a pure-genome, fully packaged AAV that contains a 2763 nt (0.84 MDa) genome is shown in Figure 2B. The MP analysis reveals a single, narrow peak at 4.60 MDa with σ = 110 kD. The chromatogram analysis determined a genome size of 2.7 knt and a total capsid of 1.14 × 10^13^ Cp/mL. The measured genome size agrees with the theoretical value, and the titer closely matches the supplier-provided value of 1 × 10^13^ Vg/mL, determined via qPCR. These results confirm that for AAV samples of high genome purity, both qPCR and SEC-MP can reliably determine their titer values.

The results shown in Figure 2 also indicate that even highly purified, homogeneous AAV samples still contain a significant amount of impurities. As shown in the chromatograms, signals at approximately 7 mL indicate the presence of AAV aggregates, and the signals in fractions eluting after 10 mL represent the small molecule contaminates.

In order to assess the reproducibility of SEC measurements, we conducted an evaluation using bovine serum albumin (BSA) as a representative analyte. During this analysis, we determined the standard deviation of the integrated peak absorbance of five independent replicates, resulting in a coefficient of variation of 0.5%. This finding emphasizes the robustness and reliability of this method.

### 3.2. SEC-MP of Empty and Full AAV Mixture Samples

When AAV samples contain only two distinct species, such as the empty capsids and fully packaged AAVs, and both of the two species exhibit homogeneity (σ ≤ 150 kDa), the chromatogram analysis can also directly yield the total capsid concentration and the %full, as described in scenario 2.

To demonstrate the validity of this analysis for the mixtures of two species, we first reanalyzed single species AAV data, shown in Figure 2, using the procedure described in scenario 2. In this case, the single species data were treated as mixtures consisting of 100% of one component and 0% of the other component. The known genome size of 839 kD was used for the chromatogram analysis. The total capsid concentrations obtained from calculations following scenario 2 (Figure 3A,B) closely match the results obtained in scenario 1. The minor differences can be attributed to the accuracy of the manual determination of the peak boundaries. Notably, the calculated %full was 0 for the empty AAV capsids, and 100% for the fully packaged sample, aligning precisely with the expected values.

Subsequently, a 1:1 volume mixture of empty and fully packaged AAV samples was prepared, and the resulting sample was analyzed using the MP-SEC method and following the procedure outlined in scenario 2. Based on the results previously obtained in scenario 1 for the empty and fully packaged AAVs, the anticipated number concentration of total capsids in the mixture should be (2.40 × 10^12^ + 1.14 × 10^13^)/2 = 6.9 × 10^12^ Cp/mL, while the expected %full should be 1.14 × 10^13^/(2.4 × 10^12^ + 1.14 × 10^13^) = 83%. The results obtained from the SEC-MP method for the mixed sample are presented in Figure 3C. The total capsid concentration obtained from the chromatogram analysis is 6.6 × 10^12^ Cp/mL, with a %full of 83%, closely matching the expected values.

### 3.3. SEC-MP of Less Pure and Heterogenous AAV Samples

In many practical situations, AAV samples exhibit heterogeneity, containing varying amounts of partially or incorrectly packaged AAVs. In the presence of these poorly resolved components, the fully packaged AAV population may not form a clearly distinguishable peak in the MP mass distribution. Consequently, accurately estimating the effective AAV titers within such heterogeneous samples poses a considerable challenge.

Using the SEC-MP method, we attempted to estimate the titers of effective, fully packaged AAVs in two less pure and heterogeneous samples. The AAV used in this study was expected to encapsidate the genome comprising 4787 nucleotides. The mass of the genome calculated from its nucleotide sequence was 1.5 MDa, while the expected total mass of the fully packaged AAV was 5.25 MDa.

The MP results of the two samples are shown in Figure 4. The mass distributions of both samples are broad, with masses of the most prominent peaks below the value expected for fully packaged AAVs. This indicates that the partially packaged virus particles are predominant, and the fully packaged AAV species are either indistinguishable within the broadly distributed mass range, or present in a very low amount. Since the total mass of the fully packaged AAV is 5.25 MDa, and as described above, a pure AAV sample with high genome purity displays an MP peak σ of less than 150 kDa, particles falling within the 300 kDa range around the 5.25 MDa value (i.e., 5.10 to 5.40 MDa) were counted as fully packaged AAVs. All particles detected by the MP within the 3 MDa to 6 MDa range were tallied as the total number of AAV particles in the sample. The ratio of these two numbers was considered as the fraction of fully packaged AAVs in the sample. As described in scenario 3, the number concentration of the total capsid particles can be derived from the chromatogram analysis. Therefore, the titer of effective, fully packaged AAVs is determined by multiplying the fraction representing the effective AAVs and the total number concentration of AAV capsids (Table 1). Finally, the titer of the effective AAV can be estimated (Table 1).

Based on the results presented in Table 1, the total capsid concentration of sample B exceeds that of sample A by over tenfold. However, both samples predominantly consist of partially packaged or empty AAVs. The SEC-MP analysis shows that the titers of effective, fully packaged AAVs in the two samples are comparable. The total capsid concentrations of the two samples determined by chromatogram analysis are within one order of magnitude of the qPCR results targeting the inverted terminal repeats (ITRs), suggesting that ITR-targeted qPCR estimates the total capsid concentration. The significantly higher titer values obtained by this type of qPCR analysis may be the result of the presence of incorrectly packaged AAVs and free DNA contaminants in the sample, as evidenced by the impurities identified in SEC experiments. Notably, the AAV titers determined via DNase-qPCR targeting the GOI closely align with the results from the SEC-MP analysis. Although DNase-qPCR targeting the GOI may not accurately determine the titer of fully packaged AAVs, as the targeted genome may be incomplete, it is a valuable method to validate the SEC-MP results using orthogonal technology.

The DNA mass values obtained in scenario 3 assume uniform distribution of DNA among all of the capsids. Similarly, the %full in this scenario represents the total amount of DNA compared to the amount present when all capsids are fully loaded. However, in relation to samples A and B analyzed here, both of the %full values lack a meaningful interpretation.

### 3.4. Minimum Amount of Materials Required in SEC-MP

The experiment time and amount of material required for a measurement are important factors in determining whether a method can be broadly applied for the characterization of AAV samples. In the case of the SEC-MP, MP only requires 10 µL of the AAV sample at a concentration of approximately 10^11^ Cp/mL, and MP data acquisition and analysis can be completed within 5 min. Therefore, the total measurement time and material consumption are primarily determined via SEC. To minimize the amount of material used in the SEC-MP method, a smaller diameter column (e.g., WTC-050S5) can be used. To test the minimum amount of sample required to obtain reliable results, different volumes of AAV samples ranging from 10 µL to 0.2 µL at approximately 10^13^ Cp/mL were analyzed (Figure 5). As shown in Table 2, with the decreasing sample size, the experimental results progressively deviate from the total capsid concentration value obtained from the analysis of the largest sample size. However, even a loading of only 0.5 µL provides acceptable results of this analysis. In summary, the SEC-MP method for AAV characterization can be performed using a minimum sample size of approximately 6 × 10^9^ AAVs, and the procedure requires a duration of 30 min for each individual sample, with the potential for recovering most of the sample material.

## 4. Discussion

AAVs are used extensively as gene delivery tools in fundamental and clinical studies; however, AAV samples often contain a significant amount of partially packaged capsids, aggregates, and other impurities such as DNA, capsid fragments, and proteins. Accurately estimating the effective quantity of AAVs in these samples is crucial for their correct and safe utilization as gene delivery tools. Therefore, the demand for a novel, fast, and cost-effective method for AAV sample characterization and quantification is evident. Previous studies have shown that the single-molecule technology, MP, is capable of determining %full, identifying partially packaged AAVs, and providing insights into sample heterogeneity [20]. However, the information obtained from MP alone is not sufficient to estimate the titers of effective AAVs. The data presented in this study show that the SEC-MP method is an efficient solution for estimating the effective AAV titers in less pure and heterogeneous AAV samples. At the same time, the SEC-MP method requires a small amount of material and a relatively short experimental time. The integration of SEC with the automated MP instrument could further reduce the lab work required for SEC-MP analysis.

AUC is another technique that can estimate effective AAV titers. The sedimentation coefficient distributions provide information about the sample heterogeneity, and the UV absorbance signals at 280 nm and 260 nm collected via AUC can be used to calculate the protein and DNA content [24]. Considering comparable resolutions of MP mass distributions and AUC sedimentation coefficient distribution obtained in AAV analyses [20], both SEC-MP and AUC should provide similar accuracy in estimating the effective titer.

However, challenges arise when differently packaged AAVs are not fully resolved in the AUC sedimentation coefficient distributions, c(s). In such cases, the features of the c(s) distributions representing the polydisperse sample will depend on the regularization parameters applied in the data analysis, potentially leading to less reliable titer estimations. In contrast to AUC c(s) analysis, the MP distributions are histograms directly enumerating virus particles based on their detected molecular weight. Since in SEC-MP the %full is calculated as the number of particles detected in the molecular weight range of interest relative to the total number of virus particles, the MP distributions provide a more reliable estimate of this value.

Another advantage of the SEC-MP method is an effective separation of small impurities that may be present in the AAV samples via size-exclusion chromatography. In contrast, it is challenging to accurately quantify both very small and very large impurities in a single AUC run. Importantly, SEC-MP requires only 1/500th of the material and 1/6th of the experimental time compared to AUC. This significant reduction in resource consumption positions the SEC-MP method as an appealing choice for the efficient and precise estimation of effective AAV titers.

The radar charts shown in Figure 6 compare five key aspects of AAV characterization technologies mentioned in this manuscript. This facilitates an intuitive comparison of SEC-MP with other currently used techniques.

In a typical MP measurement, the presence of a high concentration of cosolutes, even if they are too small to be directly detected via MP, can increase background noise in the MP image, thus increasing the lower limit of the detectable masses. To assess the influence of the presence of small molecule impurities on the MP results, both the AAV stock and the AAV fraction collected after the SEC purification were subjected to MP analysis (Appendix A). As expected, the SEC-purified AAV sample exhibited a significantly reduced level of small impurities. However, SEC purification did not substantially improve the MP data quality for AAV measurements. This can be explained by the fact that the size of the AAV particles is sufficiently large to generate strong MP signals that can be accurately measured even when small impurities increase the background noise signal levels (Appendix A).

The accuracy of the SEC-MP method is limited when characterizing heterogeneous samples. One significant source of errors is introduced by the arbitrary selection of the peak boundary positions in the chromatogram analysis. We aimed to include all signals corresponding to monomeric AAVs while avoiding including signals from impurities. However, when the peaks corresponding to impurities are not well separated from the monomeric AAV peak, the precise placement of the boundaries becomes challenging. As shown in Figure 1, peak shoulders were observed on both sides of the peak of interest, indicating the presence of aggregates and small fragments. Selecting a narrower signal integration range provides a more accurate estimation of the %full, since the signal included in the analysis corresponds to the sample fraction better isolated from impurities. However, this approach might lead to an underestimation of the total capsid concentration.

The second major source of errors originates from the selection of the MP peak of interest boundaries when the peak corresponding to the fully packaged AAVs is not well separated from the partially packaged species peaks. In this study, we arbitrarily used a 300 kDa peak boundary width positioned at the theoretical mass of the fully packaged AAV. However, the accuracy of the peak boundary placement depends both on the proximity of masses of different species, and on their respective abundances (Appendix A). This could potentially result in several-fold errors when determining the fully packaged AAV fraction size.

## 5. Conclusions

In SEC-MP, the combination of size exclusion chromatography and mass photometry provides comprehensive information regarding sample purity (SEC chromatograms) and homogeneity (MP distributions). Despite the inherent challenges associated with highly heterogeneous AAV preparations, the SEC-MP method offers several advantages compared to currently used methods. It is faster than alternative techniques and requires significantly less material. SEC-MP could also be the best approach to provide titer estimates for heterogenous samples (samples containing a large amount of empty and partially packaged AAVs) as well as AAV samples of lower purity.

With the significant advantages of SEC-MP outlined in this paper, this method emerges as the most feasible approach for obtaining reliable estimates of effective AAV titers in complex samples. Its ease of implementation further emphasizes its significance in addressing the complexities of AAV characterization.

## Figures and Tables

**Figure 1 cells-12-02264-f001:**
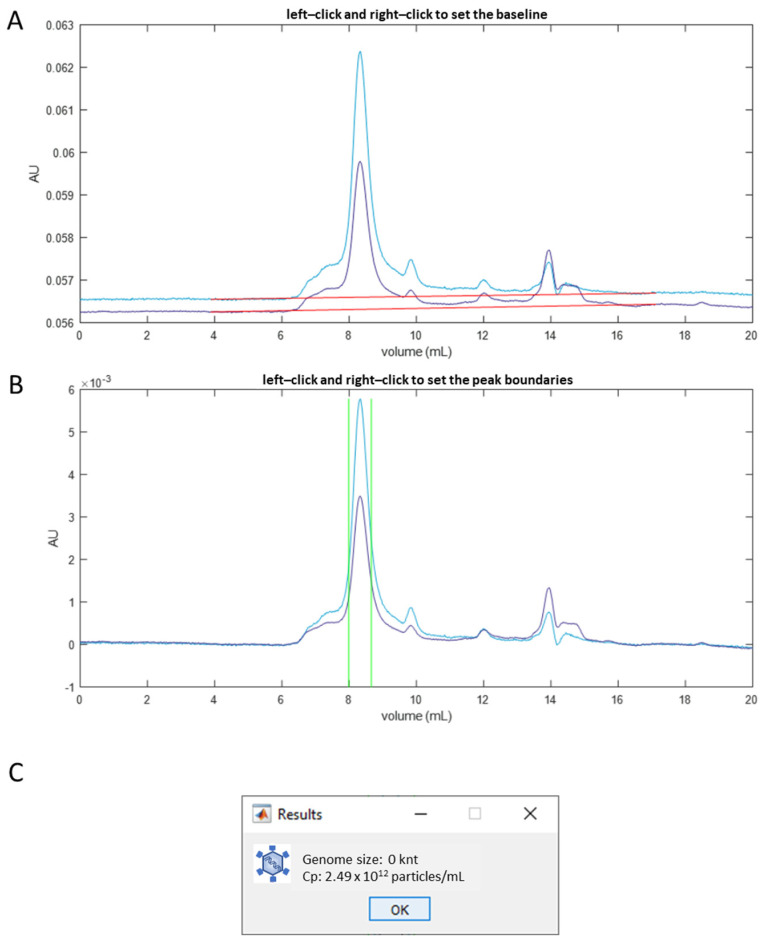
Chromatogram analysis of AAV samples. In panel (**A**,**B**), blue and purple lines represent the chromatograms of 280 nm and 260 nm absorbance, respectively. Red lines in panel (**A**) show the baselines. Panel (**B**) shows the chromatograms after baseline correction. Vertical green lines denote the peak boundaries. The output of a chromatogram analysis program implemented in MATLAB is shown in panel (**C**).

**Figure 2 cells-12-02264-f002:**
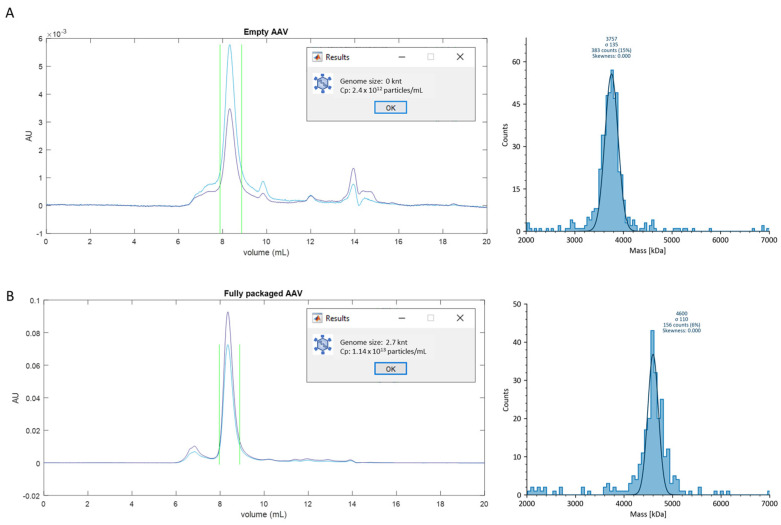
SEC−MP results of single-species AAV samples. Both empty (**A**) and fully packaged (**B**) highly pure-genome AAVs were evaluated using the SEC−MP method. The left panels show the absorbance chromatograms of AAV samples obtained with the 7.8 × 300 mm SEC column. Blue and purple lines represent the 280 nm and 260 nm absorbance signals, respectively. Green vertical lines represent the peak boundaries. The right panels show the MP results. Solid lines represent the Gaussian fits of the mass distribution histograms. The chromatogram analysis results are shown in the subpanels.

**Figure 3 cells-12-02264-f003:**
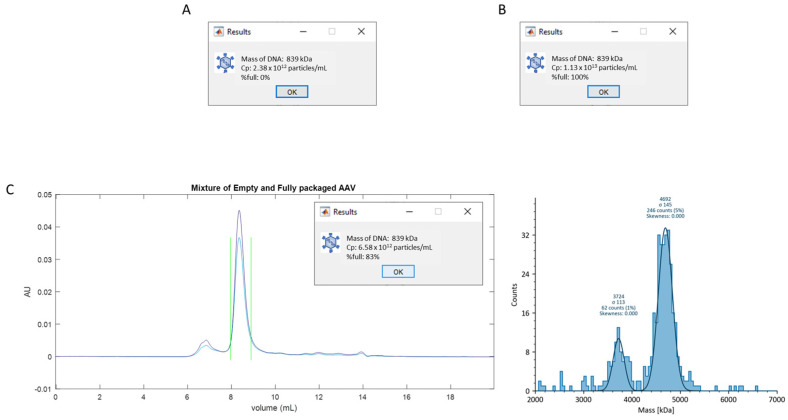
SEC −MP results of empty and fully packaged AAV mixtures. Both empty (**A**) and fully packaged (**B**) highly homogeneous AAVs were reanalyzed using the method described in scenario 2. Panel (**C**) shows the absorbance chromatograms (left) and the MP mass distribution (right) of the mixture of empty and fully packaged AAVs. Blue and purple lines in the chromatograms represent the 280 nm and 260 nm absorbance signal, respectively. Green vertical lines represent the peak boundaries. Solid blue line in MP mass distribution represents the Gaussian fit, and the chromatogram analysis results are presented in the subpanel.

**Figure 4 cells-12-02264-f004:**
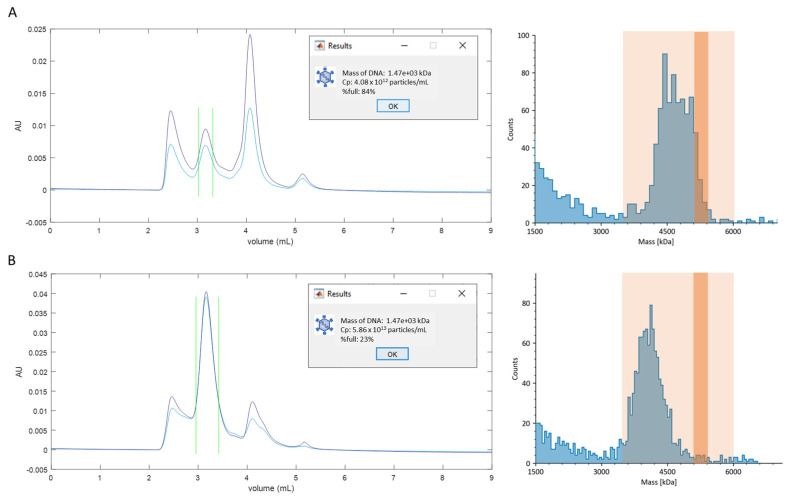
SEC−MP results of less pure and heterogeneous AAV samples. Panels (**A**,**B**) show the results for two different samples, A and B, respectively. A small-diameter, 4.6 × 300 mm SEC column (WTC−050S5) was used in the measurements. Blue and purple lines in the chromatograms represent the 280 nm and 260 nm absorbance signals, respectively. Green vertical lines represent the peak boundaries. Lighter orange shades in MP distributions represent the 3–6 MDa region and darker orange shades represent the 5.1–5.4 MDa region.

**Figure 5 cells-12-02264-f005:**
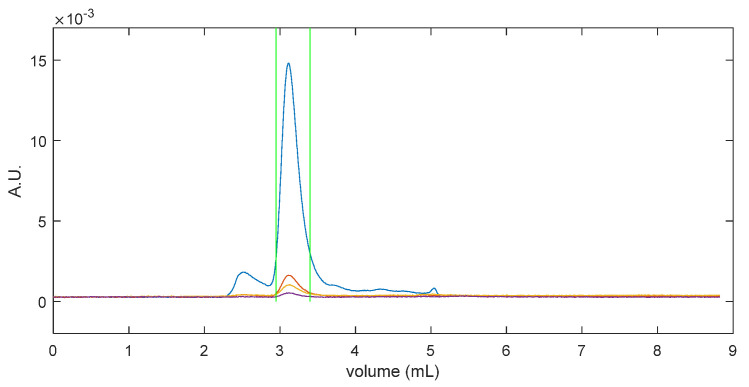
Chromatograms of UV280 absorbance signals obtained using the WTC−050S5 column. The blue, orange, yellow, and purple colors represent the sample loading volumes of 10 µL, 1 µL, 0.5 µL, and 0.2 µL, respectively. Green vertical lines represent the peak boundaries.

**Figure 6 cells-12-02264-f006:**
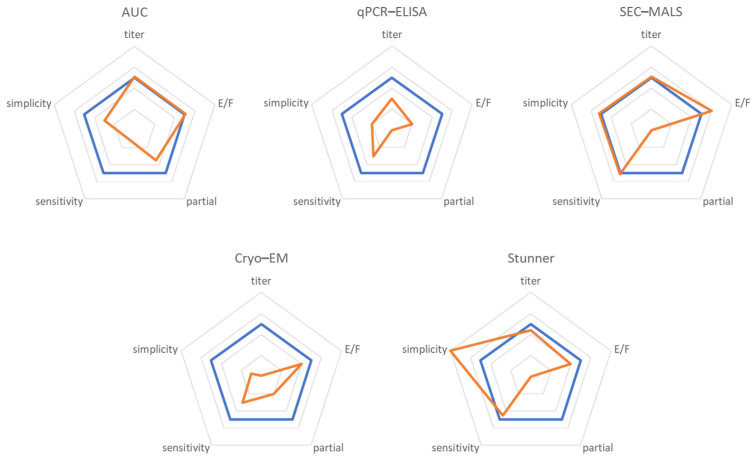
Comparative analysis of SEC-MP and other AAV characterization technologies. Five distinct technologies (orange) were qualitatively compared to SEC-MP (blue). The comparison is based on five critical attributes: titer determination (titer), where a larger value indicates a more accurate determination; %full determination in a sample containing only empty and fully packaged AAVs (E/F), where a larger value indicates a more accurate determination; %full determination and quantification of partially packaged AAVs (partial), where a larger value represents a more reliable estimation; sensitivity (sensitivity), where a larger value indicates lower material consumption; and simplicity (simplicity) of an experiment, where a larger value implies the lower experimental time and labor required.

**Table 1 cells-12-02264-t001:** Titers of less pure and heterogeneous AAV samples.

Method for Titer Determination	AAV Sample	
	A	B
Chromogram analysis	4.08 × 10^12^	5.86 × 10^13^
MP (% of full) *	8.70% ± 1.7%	1.20% ± 0.2%
SEC−MP	3.55 × 10^11^	6.95 × 10^11^
qPCR targeting ITRs **	1.55 × 10^13^	1.74 × 10^14^
qPCR targeting GOI ***	3.14 ± 0.56 × 10^11^	9.19 ± 1.43 × 10^11^

* Measurements were repeated 5 times. The average percentages of the fully packaged AAV and their standard deviations are shown in the table. ** Data obtained from the vendor. Titers were determined via qPCR, targeting the AAV inverted terminal repeats (ITRs). *** PCR assays repeated 4 times. The average titer and standard deviation are shown in the table.

**Table 2 cells-12-02264-t002:** Results of SEC-MP analysis of samples of varying size.

Loading Volume	Total (Cp/mL)	(%full)
10 µL	1.16 × 10^13^	99%
1 µL	1.06 × 10^13^	99%
0.5 µL	1.19 × 10^13^	95%
0.2 µL	7.01 × 10^12^	136%

## Data Availability

The dataset generated during the current study is available from the corresponding author upon reasonable request.

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
