# Peer review of "Size Exclusion Chromatography–Mass Photometry: A New Method for Adeno-Associated Virus Product Characterization"

_cells, 2023, doi:10.3390/cells12182264_

Round 1
Reviewer 1 Report
This paper describes the development of an SEC-MP method for analyzing the empty, full, and partial capsids in AAV preparations.
The work appears to be well-designed and executed and of value to the field. The writing and presentation is generally clear, with a few points noted below that would improve the clarity and readability.
46 to110: methods
A further step for UV - line 98-105 suggest you also mention UV absorbance to estimate empty/full; see Porterfield et. al. - the 2 UV signals at 260 and 280 nm can be used to determine the total nucleic acid and total protein in the sample and this can then be converted to 'equivalent' (average bulk) overall empty and full and therefore an apparent %full, not just total protein and total nucleic acid. This has a concentration limitation though, plus is an average/equivalent so would not discriminate partials.
suggest add info about ddPCR, which is more accurate than qPCR. External DNA can be digested as part of the method, eliminating that source of error.
Line 89/90: ddPCR would be the titer measure used for studies in any case.
Line 81-91: if you could please describe the concentration needed for bulk MP measurements - is there a low limit etc.
Use of terminology and abbreviations is not clear -
Please define Vg and all other symbols presented in equations. I assume this is vector genomes in full capsids? And Vg/Cp is the % full?
lines 155 to 176: the math you describe is not clear - you talk about taking ratios etc but without providing the mathematical symbols of the variables you are describing. For example " ...determined by calculating the ratio of these two numbers." - this is very unclear and vague and requires effort from the reader to interpret your text into equations and relationships.
To clear all this up I suggest you define variables/measurements and provide an equation for each analysis you are describing.
The peaks at 10 mL may (I would venture they likely are) be excipient species, not impurities.
Line 257 - why not do a series of mixture ratios of empty and full, rather than just a 1:1 mixture?
Table 1. what is CF and PP? Do these acronyms have a meaning? for the entire paper - please define acronyms and symbols more clearly.
Figure 6 - E/F is not something I have seen presented often. The value for % full ( Vg/Cp) seems much more helpful rather than E/F, a ratio of the two.
so % full = Vg/Cp = Vg/(Vg + Cp) ?
and Vg = F?
it seems you are using two different symbols for the same thing?
how would one use the ratio E/F? it doesn't seem that useful to me compared to %full.
Figure 6. I would say that ddPCR is the standard and best method for titer measurements of vector genomes per mL.
The radar plot in Figure 6 is not very easy to read or helpful. I suggest you present this in a different way.
I would say ddPCR is best for titer and AUC is the standard for empty/full/partials for final purified material, but I don't see people using it to estimate titer.
(note - I would define titer and 'vector genomes in intact full capsids' - i.e. the amount of AAV that can actually transfect cells. I am not sure if you are using the same or a different defintion?
In my experience, for a pre-clinical study the vector genome concentration (ie. full particles/mL) would be the measure used for dosing. The total capsids/mL, of all types would be a secondary measure - most often simply presented as %full rather than listing out the total capsids per mL.
Reviewer 2 Report
The authors are tackling a crucial issue, highlighting the necessity of characterizing AAV preparations used in the different studies. This is vital for ensuring reproducible outcomes across different labs, as AAV preparations often contain impurities and exhibit varying proportions of empty capsids and capsids with incomplete genomes. The method proposed by the authors holds the potential to effectively address this challenge. Nevertheless, there is a need for enhancements in several aspects to effectively demonstrate the potential and robustness of the presented methodology.
Initially, it remains unclear whether the authors conducted the experiments with biological replicates to ensure the reproducibility of the presented findings through multiple measurements. So it would be very valuable if the authors could provide this information.
Conversely, the most significant application of this methodology would be its utilization with heterogeneous AAV samples, given that these are the most commonly employed preparations. In this context, employing a genome size of 4787 nucleotides seems to be well within the capacity limit of an AAV, and concerns about "challenges and inefficiencies in the assembly process" may not hold true.
A notable deficiency in this study is the absence of data validation when characterizing both heterogeneous samples (CF and PP). To establish the accuracy of the presented data, it is imperative that the authors provide a more comprehensive description of the two samples employed. This should encompass a comprehensive characterization of the samples using well-established techniques (such as AUC), demonstrating the potential of the presented methodology by illustrating that the acquired outcomes align with expectations and offering evidence that highlights the enhanced aspects facilitated by this methodology.
In their conclusions, the authors acknowledge the complexities posed by heterogeneous samples while asserting that their methodology represents a superior approach for estimating effective AAV titers. In order to substantiate this conclusion, it is imperative that the authors address the aforementioned points by providing the requested data.
Round 2
Reviewer 2 Report
The responses provided by the authors have effectively addressed most of the concerns, leading me to conclude that the manuscript can be accepted.